# Multi-Predictor Fusion: Combining Learning-based and Rule-based Trajectory Predictors

**Sushant Veer[1], Apoorva Sharma[1], Marco Pavone[1,2]**
[1]NVIDIA Research, [2]Stanford University
{sveer,apoorvas,mpavone}@nvidia.com

**Abstract:** Trajectory prediction modules are key enablers for safe and efficient planning of autonomous vehicles (AVs), particularly in highly interactive traffic scenarios. Recently, learning-based trajectory predictors have experienced considerable success in providing state-of-the-art performance due to their ability to learn multimodal behaviors of other agents from data. In this paper, we present an algorithm called multi-predictor fusion (MPF) that augments the performance of learning-based predictors by imbuing them with motion planners that are tasked with satisfying logic-based rules. MPF probabilistically combines learning- and rule-based predictors by *mixing* trajectories from both standalone predictors in accordance with a belief distribution that reflects the online performance of each predictor. In our results, we show that MPF outperforms the two standalone predictors on various metrics and delivers the most consistent performance.

**Keywords:** trajectory prediction, rule-based planning

## 1 Introduction

The behavior of a traffic agent can depend on a very large number of factors such as the desired navigation goal, personal driving style, local driving customs, etc., therefore, predicting future trajectories of traffic agents is a fundamentally challenging task. Deep learning-based approaches [1, 2, 3] have emerged as front runners in this task due to their ability to extract patterns from complex high-dimensional data and generate multimodal predictions. However, improving the interpretability of these predictors and endowing them with ability to reason about the traffic law and nuanced traffic scenarios remains an open challenge. Forcing the learned predictor to strictly conform with a specific set of logic rules can make the predictions brittle and sensitive to the completeness of the rules; for instance, when an agent violates the traffic rules, a strict traffic-law abiding predictor would forecast incorrect trajectories. In this paper, we present an approach to bake rules in the predictor while retaining the flexibility afforded by a data-driven only approach and demonstrate its ability to perform well on various real-world trajectory prediction datasets and metrics.

We introduce multi-predictor fusion (MPF) that combines learning-based trajectory predictors with planning-based predictors using Bayesian belief updates. We maintain a belief distribution over the standalone predictors. The probability assigned to each predictor by the belief distribution is updated by monitoring their individual performance online; the belief probability mass is iteratively shifted towards the better-performing predictor. Through this approach, MPF is able to relax the use of rules if the data-driven approach is performing better, and vice-versa.

MPF provides various benefits: First, we observe that learning-based predictors and rule-based predictors perform well on complementary data regimes. Learning-based predictors perform better in scenarios where multiple future outcomes are feasible due to their ability to reason about multi-modality. On the other hand, rule-based planners can incorporate traffic laws and perform better when the agent's intended goal is unambiguous or the scene is out-of-distribution. The scatter plot in Fig. 1(a) illustrates the complementary performance on different data: the average displacement error (ADE) for a large number of scenes lies far away from the identity line, suggesting that one of the two predictors is performing much better on them. Therefore, soft switching between the

7th Conference on Robot Learning (CoRL 2023), Atlanta, USA.

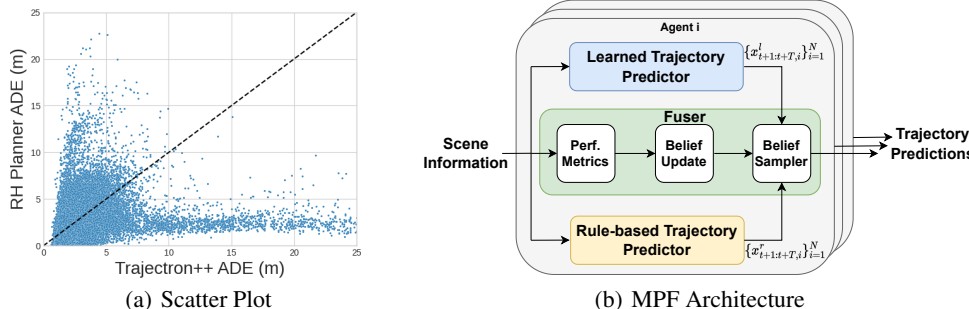

|   |   |
|---|---|
| (a) Scatter Plot | (b) MPF Architecture |

Figure 1: Motivation for predictor fusion and the architecture for multi-predictor fusion. (a) The scatter plot depicts the Average Displacement Error (ADE) for each agent-centric prediction in the nuPlan-mini dataset for learned (Trajectron++ [1]) and rule-hierarchy (RH) planner based predictor. (b) Architecture of MPF predictor.

predictors via the belief distribution can improve the overall prediction performance, and, in fact it does, as we will discuss later in Section 4. Second, MPF is highly modular and can be used with any off-the-shelf predictors[1] without re-training them. This ability to "plug-and-play" predictors facilitates easier updates to the prediction module, e.g., adaptation to traffic laws in different geographical regions. Finally, MPF is backpropagatable which facilitates end-to-end data-driven training / fine tuning of the AV stack.

MPF includes three key components: a learned trajectory predictor, a rule-based trajectory predictor, and a fuser (see Fig. 1(b)). We can use any learned trajectory predictor in MPF. For the rule-based predictor we design a motion planner that plans for the agent that we are predicting for according to a hierarchy of rules [4, 5, 6]. Rule hierarchies mirror the tendency of human drivers to relax lower priority rules (e.g., speed limit) in favor of higher priority rules (e.g., collision avoidance),which is the key to realizing realistic driving behaviors [7]. Furthermore, they can be encoded as a scalar reward that can be used for fast online planning [6]. The fuser compares the performance of all predictors according to some task-specific metric (e.g., displacement error, planning cost, etc.) and maintains a belief distribution on the standalone predictors using $\eta$-generalized Bayes updates [8].

**Statement of Contributions.** The contributions of this paper are threefold: (i) We present a new method for fusing learned and rule-based trajectory predictors such that they operate within the data regime that they are proficient at. (ii) We develop a rule-based trajectory predictor using rule hierarchies [6] that achieves a level of performance comparable to Trajectron++ [1]. (iii) Finally, we demonstrate that MPF delivers more consistent performance across various prediction metrics and on multiple trajectory prediction datasets than the learning- and rule-based predictors alone.

## 2 Related Works

**Rule-based Trajectory Prediction.** Human behavior typically conforms to a set of broad rules, customs, and norms. Thus, many works on trajectory prediction have relied on building rule-based models, with the underlying assumption that agents will act in compliance with a specified rule set. Rules can range from simple heuristics [9], to enforcing physical constraints, or rules encoding semantic traffic rules and norms, such as lane following [10]. We refer to [11] for a thorough survey on such methods. The prediction quality of such models is generally limited by the rigidity or incompleteness of rulesets. While recent work has demonstrated that appropriately designed rules can be competitive in structured environments [7], such models still struggle in corner cases.

**Learning-based Trajectory Prediction.** These challenges has led to a proliferation of learning-based approaches which aim to sidestep these issues by learning directly from datasets of observed agent behavior. In particular, many deep learning architectures have been developed for this task, varying in how they encode temporal sequences of data (e.g. using recurrent neural network (RNN) [12, 13] or Transformer architectures [3]), how they reason about agent interactions (e.g. using pooling operations [12], graph neural networks (GNNs) [2, 1], or attention mechanisms [3]), and how

---

[1]MPF can be used for any number and type of predictors (learning- or rule-based); however, in this paper, we will limit our scope to fusing a learning- and a rule-based predictor.

they represent their predictions (e.g. a single deterministic prediction [12], a probability distribution over future trajectories [1, 14], or a set of samples [15]). Much of the focus of this research area has been in ensuring that these models are able to condition on all available contextual cues, such as lane geometry and semantic map information. Advances in model architectures paired with the availability of high-quality large-scale datasets [16, 17, 18, 19] have led to impressive performance, especially when evaluated on held-out test splits.

**Fusing Learned and Rule-based Trajectory Prediction.** Pure learning-based approaches can struggle in corner cases, producing predictions that fail to obey common-sense road rules, e.g., drifting outside the road boundaries. To address such issues, recent work has considered fusing some rule-driven structure into learning-based trajectory prediction models. One style of approach is to first generate a set of candidate plans that satisfy kinematic feasibility and semantic traffic rules via explicit rules-based motion planning, before using a learned model to subselect from these fixed "anchors" to produce predictions [20, 21]. Alternatively, [22] consider applying rules, encoded in the form of signal-temporal-logic specifications, as a post-processing step to "correct" the predictions made by a learning-based model. These approaches connect learned prediction and rule-based reasoning in series; in contrast, we propose an architecture which puts the two in parallel, dynamically switching between the two on a per-agent basis. As such, it is trivial to alter one component (e.g. adjusting the rules for the rule-based model) without retraining the learning-based components. The closest work in the literature to our parallel architecture is [23]. However, [23] uses a different approach than us to fuse the learning- and planning-based predictors. Furthermore, unlike us, [23] assumes a priori availability of the agent's ground-truth intent; we present a thorough comparison with the fusion approach in [23, Section II-F] in Appendix C provided in the supplementary.

**Recursive Bayesian Multi-Model Filtering.** Our approach builds on core ideas of recursive Bayesian filtering, a cornerstone of probabilistic robotics. In particular, our approach is similar in spirit to early work in human trajectory forecasting which leveraged an interacting multiple models (IMM) filter [24, 25] to fuse predictions between basic single-rule models (constant velocity, constant acceleration, constant turning) [26]. Similar ideas have been applied for vehicle prediction, fusing simple kinodynamics based predictions with semantic map-based predictions (lane following, lane changing, etc.) [27]. In this work, we take a similar approach to fuse predictions from a learning-based model with a novel, more-expressive, rule-based prediction model.

## 3 Multi-Predictor Fusion

In this section, we will discuss the key building blocks of MPF, shown in Fig. 1(b). First, we define our notation: Let $x \in \mathcal{X}$ be the state of the traffic agent for which we are predicting and $y \in \mathcal{Y}$ be the joint state of all traffic agents in the scene. Let $x_{t:t+\tau}$ and $y_{t:t+\tau}$ be the discrete-time state trajectories over some time duration from $t$ to $t + \tau$ that lie in the space denoted by $\mathcal{X}'$ and $\mathcal{Y}'$, respectively. The scene map is described by a vector $m$ that lies in the space of map features $\mathcal{M}$.

### 3.1 Learning-Based Trajectory Predictor

Learning-based trajectory predictors, denoted by $l$, typically take the form of generative networks that produce a probability distribution $\mathbb{P}(x_{t+1:t+T}|y_{t-H:t}, m, l)$ on future trajectories over a horizon of length $T$, conditioned on the historical state trajectories of all agents in the scene and the map. For the sake of exposition, we have included the map information in the learning-based predictor, however, this is not strictly needed; learned predictors that do not use map information can also be used in our approach. At each prediction run, we sample $N$ trajectories from the learned predictor, i.e., $\{x^l_{t+1:t+T,i}\}_{i=1}^N \sim \mathbb{P}(x_{t+1:t+T}|y_{t-H:t}, m, l)$, and pass them to the fuser, as shown in Fig. 1(b).

### 3.2 Rule-based Trajectory Predictor

To complement the learned predictor, we propose a rule-based trajectory prediction model which predicts agent behavior based on a specified traffic code. Importantly, rather than forcing agents to satisfy a strict set of rules, we specify this code in the form of a rule-hierarchy, allowing agents to relax satisfaction of less important rules (e.g., speed limit) if needed to ensure satisfaction of more important rules (e.g. collision avoidance). This flexibility allows for handling edge-case scenarios

in a scalable manner due to its ability to naturally adapt to exceptional scenarios without additional supervision and has been shown to produce realistic driving behaviors [7].

**Rules.** We express the rules that we want a vehicle to satisfy as boolean expressions $\phi : \mathcal{X}' \times \mathcal{Y} \times \mathcal{M} \rightarrow \{\texttt{True}, \texttt{False}\}$ in the form of Signal Temporal Logic (STL) [28] using STLCG [29]. Each STL rule is equipped with a robustness metric $\rho : \mathcal{X}' \times \mathcal{Y} \times \mathcal{M} \rightarrow \mathbb{R}$ which returns positive values if the rule is satisfied and negative values otherwise; larger positive robustness values indicate greater satisfaction of the rule while smaller negative values indicate greater violation.

**Rule Hierarchy.** A rule hierarchy $\varphi := \{\phi_i\}_{i=1}^n$ is defined as a sequence of rules indexed in decreasing order of importance. The robustness vector $\rho \in \mathbb{R}^n$ of the rule hierarchy is defined as the vector $\rho := (\rho_1, \cdots, \rho_n)$ of the individual robustness of each rule $\phi_i$. The rule hierarchy induces a ranking of trajectories in accordance with the importance of the rules they satisfy; trajectories that satisfy more important rules receive a superior rank than those that satisfy less important rules. For instance, for a 2-rule hierarchy $\{\phi_i\}_{i=1}^2$, trajectories that: satisfy both rules have rank 1, satisfy $\phi_1$ but violate $\phi_2$ have rank 2, satisfy $\phi_2$ but violate $\phi_1$ have rank 3, and violate both rules have rank 4; see [6, Defintion 1] for a formal definition of the rank of a trajectory.

**Planner.** The planner first chooses the nearest lane to the agent (in terms of the Cartesian distance as well as the orientation) as the lane for the agent to follow; the desired lane is denoted by the black dashed line in Figs. 2(a) and 2(b). The rule hierarchy we use in this paper is presented in Fig. 2(c) which contains four rules in decreasing importance: collision avoidance, follow the center polyline of the lane to be followed, orient along the center polyline of the lane to be followed, and speed limit. The planner is tasked with selecting trajectories with the highest possible rank. This can be achieved efficiently by leveraging the rank-preserving reward function $R : \rho \mapsto R(\rho) \in \mathbb{R}$ of a rule hierarchy that embodies the property: trajectories with a higher rank receive a higher reward than trajectories with a lower rank. We borrow the rank-preserving reward function from [6, Theorem 1]:

$$R(\rho) = \sum_{i=1}^n \left( a^{n-i+1}\text{step}(\rho_i) + \frac{1}{n}\rho_i \right), \tag{1}$$

where $a > 2$. We generate a trajectory tree with $K$ branches $\{x_{t+1,:t+T}^1, \cdots, x_{t+1:t+T}^K\}$ using splines [14] in the same manner as described in [30] and compute the rank-preserving rewards $\{R_1, \cdots, R_K\}$ for each trajectory; trajectory tree generation and reward computation are parallelized on the GPU. The trajectory tree and the rewards of the branches can be visualized in Figs. 2(a) and 2(b); trajectory tree branches with warmer colors receive a higher reward while branches with cooler colors receive a lower reward. All vehicles, other than the one for which we are predicting, are assumed to move with a constant velocity over the prediction horizon for computing the reward. We adopt the constant-velocity prediction model for two main reasons: (i) Its computational efficiency allows batching predictions for all traffic agents, enabling fast online runtimes; see Section 4. (ii) The constant-velocity predictor has been shown to perform reasonably well [31].

**Trajectory Prediction.** Rather than choosing the trajectory with the highest reward, we transform the trajectory tree to a discrete Boltzmann distribution $\mathbb{P}(x_{t+1:t+T}|y_{t-H:t}, m, r)$ by viewing the rewards as the negative of the Boltzmann energy as follows:

$$p_i = \frac{\exp(R_i/\zeta)}{\sum_{i=1}^K \exp(R_i/\zeta)}, \tag{2}$$

where $\zeta > 0$ is the temperature of the Boltzmann distribution, controlling the degree of optimality we expect from real-world agents with respect to the chosen rule hierarchy. At each prediction run, we sample $N$ trajectories from the rule-hierarchy predictor, i.e., $\{x_{t+1:t+T,i}^r\}_{i=1}^N \sim \mathbb{P}(x_{t+1:t+T}|y_{t-H:t}, m, r)$, and pass them to the fuser, as shown in Fig. 1(b).

### 3.3 Fuser

To fuse the learned trajectory predictor with the rule-based predictor, we propose a recursive Bayesian filtering scheme, similar to the interacting multiple model (IMM) filter. At a high level, our approach maintains a belief over which of the learned and rule-based models best fits a particular agent's behavior, and fuses their predictions according to this belief. During runtime, as we observe

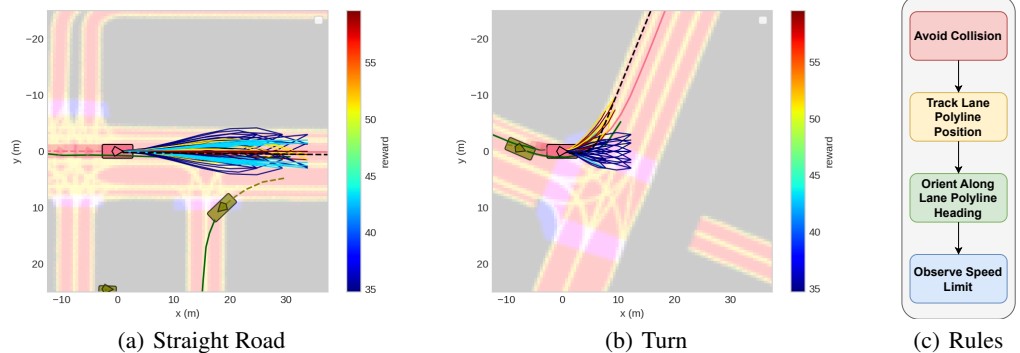

Figure 2: Demonstration of the rule-hierarchy planner. In the first two figures, the pink vehicle is the agent we are planning for. **(a)** Planning on a straight road. **(b)** Planning on a turn. **(c)** Rule hierarchy used in this paper.

agent behavior, we measure the performance of each individual model, and use this estimate to recursively update this belief. In the rest of this section, we will discuss the mechanics of the belief update and the building blocks of the fuser (shown in Fig. 1(b)) in more detail.

### 3.3.1 Performance Metric

The performance metric $\Gamma : \mathcal{X}^N \times \mathcal{X} \to [0, \infty)$ can be any function that assesses how well the observed state of the agent $x_{t+1}$ conforms to the *past* predictions $\{x_{t+1,i}^l\}_{i=1}^N$ and $\{x_{t+1,i}^r\}_{i=1}^N$. This block, effectively, serves as a run-time monitor on the "quality" of both the individual predictors. Examples of performance metrics include average displacement error, final displacement error, likelihood of a kernel density estimate, downstream planning cost, or any other task-specific metric.

### 3.3.2 Belief Update

We follow a Bayesian approach to update the belief distribution. At the beginning of each episode we choose a prior belief $b_0$, e.g., an uninformed prior $(0.5, 0.5)$, that reflects our initial knowledge about the correctness of the two models. Each belief update consists of two parts: (i) the observation step, and (ii) the mixing step. In the observation step, we collect the observed agent state $x_{t+1}$, and use Bayes rule to obtain the conditional probabilities

$$b_{t+1}^l = \mathbb{P}(l \mid x_{t+1}, y_{t-H:t}, m) = \frac{\mathbb{P}(x_{t+1} \mid y_{t-H:t}, m, l)b_t^l}{\mathbb{P}(x_{t+1})}, \tag{3}$$

$$b_{t+1}^r = \mathbb{P}(r \mid x_{t+1}, y_{t-H:t}, m) = \frac{\mathbb{P}(x_{t+1} \mid y_{t-H:t}, m, r)b_t^r}{\mathbb{P}(x_{t+1})}. \tag{4}$$

Using the above, we can express $b_{t+1}^l$ in terms of $b_{t+1}^r$ as follows:

$$b_{t+1}^l = \alpha b_{t+1}^r, \qquad \text{where} \quad \alpha := \left( \frac{\mathbb{P}(x_{t+1} \mid y_{t-H:t}, m, l)b_t^l}{\mathbb{P}(x_{t+1} \mid y_{t-H:t}, m, r)b_t^r} \right). \tag{5}$$

is the ratio of the likelihoods. In practice, we only obtain samples from each trajectory predictor, and cannot directly evaluate a likelihood of the trajectories[2]. Instead, we can estimate the likelihoods in $\alpha$ by using the performance metrics $\Gamma^l := \Gamma(\{x_{t+1,i}^l\}_{i=1}^N, x_{t+1})$ and $\Gamma^r := \Gamma(\{x_{t+1,i}^r\}_{i=1}^N, x_{t+1})$ as the likelihood of observing $x_{t+1}$, i.e., $\alpha = \Gamma^l/\Gamma^r$. Intuitively, a high performance metric corresponds to the predictor doing well at predicting the agent trajectory, and hence a high performance metric should correspond to a high likelihood of observing the agent's trajectory. Mathematically, the only condition we need to ensure about $\Gamma$ is that it should be integrable over its domain. This can always be achieved for positive performance metrics (e.g., ADE) by defining $\Gamma$ as the composition of the performance metric with a negative exponential. Rather than using the standard Bayes update directly, we use the $\eta$-generalized Bayes update [8] here which introduces a scalar parameter $\eta \in (0, 1)$ in (5):

$$b_{t+1}^l = \alpha^\eta b_{t+1}^r. \tag{6}$$

---

[2]We found that non-parametric likelihood estimation, e.g. using a kernel density estimation, led to numerical instability

The parameter $\eta$ is called the learning rate and controls how fast the belief is updated on encountering new "evidence", i.e., $x_{t+1}$. Choosing an $\eta < 1$ has been shown to produce better performance when the probabilistic mechanism that generates the evidence is inconsistent with all models [8]; indeed, this is likely the case in real-world trajectory predictions, because no learned or rule-based model can describe *all* real-world agent behavior.

After factoring in the observation, we next account for the chance that the optimal model for a particular agent might switch, e.g. if an agent switches their behavior. To account for this, we update the belief by mixing it with the prior, analogous to the mixing step in an IMM. Thus, the full belief update which we perform is given by:

$$b_{t+1} = (1 - \gamma) \begin{bmatrix} \frac{\alpha^\eta}{1+\alpha^\eta} \\ \frac{1}{1+\alpha^\eta} \end{bmatrix} + \gamma b_0, \tag{7}$$

where $\gamma \in (0, 1)$ is a small positive constant close to zero (e.g., 0.1) representing the chance that the agent behavior switches at any given timestep. Note that this corresponds to a geometric distribution over the duration that a given model describes agent behavior; the mean time between switching corresponds to $1/\gamma$.

### 3.3.3 Belief Sampler

Finally, to make predictions, we sample predictions from both models according to the current belief. As shown in Fig. 1(b), the belief sampler receives a set of $N$ trajectories from each model, i.e., $\{x_{t+1:t+T,i}^l\}_{i=1}^N \sim \mathbb{P}(x_{t+1:t+T}|y_{t-H:t}, m, l)$ and $\{x_{t+1:t+T,i}^r\}_{i=1}^N \sim \mathbb{P}(x_{t+1:t+T}|y_{t-H:t}, m, r)$. The belief sampler makes $N$ draws from the belief distribution $b$ to identify which predictor to sample from. Let the number of draws from the learned predictor be $N^l$ and the number of draws from the rule-based predictor be $N^r$, where $N^l + N^r = N$. Then, the belief sampler draws $N^l$ trajectories from $\{x_{t+1:t+T,i}^l\}_{i=1}^N$ and $N^r$ trajectories from $\{x_{t+1:t+T,i}^r\}_{i=1}^N$ to construct the predictions of MPF. These trajectory can either be re-sampled uniformly or we can choose the first $N^l$ or $N^r$ trajectories from $\{x_{t+1:t+T,i}^l\}_{i=1}^N$ or $\{x_{t+1:t+T,i}^r\}_{i=1}^N$, respectively, since the trajectories drawn from the predictors are independent identically distributed (i.i.d.); we use the latter approach.

A complete algorithm for MPF is provided in Algorithm 1 in Appendix A.

## 4 Experimental Evaluation

In this section, we demonstrate MPF's ability to deliver consistent performance across prediction metrics and datasets. All experiments were conducted on a desktop computer with an `AMD Threadripper Pro 3975WX` CPU and an `NVIDIA RTX 3090` GPU.

**Datasets.** We test MPF on three AV datasets: nuPlan-mini [32], nuScenes [18], and Lyft [17]. In particular, we use the validation splits of each of these datasets. For nuPlan-mini and nuScenes we use the entire validation dataset while for Lyft we use $10\%$ of the validation dataset. The data is loaded in an agent-centric manner using `trajdata` [19]. The number of prediction scenes (scene information for predicting the trajectory of an agent) and episodes (continuous sequence of agent trajectory rollout) for each dataset are provided in Table 2 in Appendix B.2.

**Predictors.** In the results, we compare Trajectron++ [1], the rule hierarchy (RH) predictor that we developed in Section 3.2, and MPF that probabilistically fuses both the former models using the Bayesian belief approach discussed in Section 3.3.2.

**Metrics.** For each frame and agent in the dataset, we sample $N$ trajectory predictions from each predictor denoted by $\{x_{t+1:t+T,i}^p\}_{i=1}^N$ (where the superscript $p$ denotes the predictor from which the trajectories are sampled). Let $\{x_{t+1:t+T}\}_{i=1}^T$ be the ground truth future trajectory of the agent for which we are predicting. We compute the following metrics: (i) Average Displacement Error (ADE) is average of the mean distance error across all $N$ predicted trajectories and the ground truth future trajectory: $\frac{1}{NT} \sum_{i=1}^N \sum_{\tau=1}^T \|x_{t+\tau,i}^p - x_{t+\tau}\|$. (ii) Final Displacement Error (FDE) is average of the distance error between the last time step of all $N$ predicted trajectories and the last time-step of the ground truth future trajectory: $\frac{1}{N} \sum_{i=1}^N \|x_{t+T,i}^p - x_{t+T}\|$. (iii) minADE is the smallest mean distance error across all $N$ predicted trajectories and the ground truth future trajectory: $\min_{i=1,\cdots,N} \frac{1}{T} \sum_{\tau=1}^T \|x_{t+\tau,i}^p - x_{t+\tau}\|$. (iv) minFDE is the smallest distance error between the last

| | Mean (m) | | | | CVaR$_{0.1}$ (m) | | | | Mean Diff From |
|---|---|---|---|---|---|---|---|---|---|
| Predictor | ADE | FDE | minADE | minFDE | ADE | FDE | minADE | minFDE | Best (MDB) (%) |
| | | | | | **NuPlan mini** | | | | |
| Traj++ | $2.38 \pm 0.01$ | $5.31 \pm 0.01$ | $0.87 \pm 0.01$ | $1.07 \pm 0.01$ | $5.42 \pm 0.07$ | $11.10 \pm 0.10$ | $3.05 \pm 0.07$ | $4.45 \pm 0.10$ | 39.64 |
| RH Pred (ours) | $\mathbf{1.66 \pm 0.01}$ | $\mathbf{3.87 \pm 0.02}$ | $0.96 \pm 0.01$ | $2.12 \pm 0.02$ | $5.05 \pm 0.04$ | $11.74 \pm 0.08$ | $3.94 \pm 0.04$ | $9.10 \pm 0.09$ | 70.96 |
| MPF (ours) | $1.95 \pm 0.01$ | $4.53 \pm 0.02$ | $\mathbf{0.54 \pm 0.00}$ | $\mathbf{0.80 \pm 0.01}$ | $\mathbf{4.34 \pm 0.02}$ | $\mathbf{10.04 \pm 0.06}$ | $\mathbf{1.74 \pm 0.01}$ | $\mathbf{3.41 \pm 0.03}$ | **4.36** |
| | | | | | **NuScenes** | | | | |
| Traj++ | $2.40 \pm 0.01$ | $5.48 \pm 0.02$ | $0.81 \pm 0.00$ | $1.06 \pm 0.01$ | $\mathbf{4.56 \pm 0.04}$ | $\mathbf{10.08 \pm 0.07}$ | $2.16 \pm 0.03$ | $\mathbf{3.49 \pm 0.05}$ | 18.17 |
| RH Pred (ours) | $\mathbf{1.82 \pm 0.01}$ | $\mathbf{4.24 \pm 0.03}$ | $0.95 \pm 0.01$ | $2.05 \pm 0.02$ | $5.32 \pm 0.05$ | $11.76 \pm 0.11$ | $4.07 \pm 0.05$ | $9.11 \pm 0.11$ | 64.68 |
| MPF (ours) | $2.09 \pm 0.01$ | $4.85 \pm 0.02$ | $\mathbf{0.54 \pm 0.01}$ | $\mathbf{0.83 \pm 0.01}$ | $4.70 \pm 0.04$ | $10.34 \pm 0.08$ | $\mathbf{2.04 \pm 0.03}$ | $3.73 \pm 0.05$ | **5.17** |
| | | | | | **Lyft** | | | | |
| Traj++ | $2.78 \pm 0.00$ | $6.04 \pm 0.01$ | $1.05 \pm 0.00$ | $1.30 \pm 0.00$ | $\mathbf{6.23 \pm 0.01}$ | $\mathbf{13.20 \pm 0.03}$ | $3.12 \pm 0.01$ | $\mathbf{5.07 \pm 0.02}$ | 8.44 |
| RH Pred (ours) | $\mathbf{2.36 \pm 0.01}$ | $\mathbf{5.36 \pm 0.01}$ | $1.53 \pm 0.00$ | $3.30 \pm 0.01$ | $7.94 \pm 0.02$ | $18.23 \pm 0.05$ | $6.54 \pm 0.02$ | $14.93 \pm 0.06$ | 79.09 |
| MPF (ours) | $2.55 \pm 0.00$ | $5.69 \pm 0.01$ | $\mathbf{0.85 \pm 0.00}$ | $\mathbf{1.25 \pm 0.00}$ | $6.63 \pm 0.02$ | $14.74 \pm 0.03$ | $\mathbf{2.87 \pm 0.01}$ | $5.18 \pm 0.02$ | **4.28** |

Table 1: Results for 4 second prediction horizon with $N = 20$ rounded to the nearest second decimal.

time step of all $N$ predicted trajectories and the last time-step of the ground truth future trajectory: $\min_{i=1,\cdots,N} \|x^p_{t+T,i} - x_{t+T}\|$. For all the above metrics, we report their mean and the 0.1-level conditional value-at-risk (CVaR) [33, 34] across all prediction scenes. CVaR$_{0.1}$ is the mean of the top-10% tail of the histogram for each metric and provides insights about the "tail" performance.

The last metric we present is the Mean Difference from Best (MDB) which combines all the above metrics to provide a comparative performance measure. MDB computes the average percentage difference between a predictor's performance on a metric from that of the best performing predictor on the same metric. Let there be $J$ metrics for each predictor $p$, denoted by $\mu^p_j$, and let $\mu^*_j$ be the metric value for the best performing predictor on that metric. Then, MDB is computed as: $\frac{1}{J} \sum_{j=1}^{J} \frac{\mu^p_j - \mu^*_j}{\mu^*_j} \times 100$. If a predictor is the best at all metrics, then it will have an MDB of 0%. We report MDB across all eight metrics discussed above, i.e., the mean and CVaR$_{0.1}$ of ADE, minADE, FDE, and minFDE.

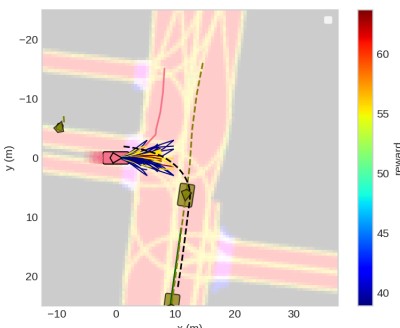

Figure 3: Incorrect lane choice.

**Results and Discussion.** We report the performance of our predictors on all datasets in Table 1 along with 95% confidence intervals. MPF performs well on various metrics across all datasets. In particular, MPF demonstrates the lowest MDB across all datasets, highlighting its *consistency* across all metrics: Low MDB indicates that even if MPF is not the best predictor on a metric, it is at least close to the best performing predictor. For example, MPF does not have the best Mean ADE and FDE in all the three tables, but its performance is closer to RH predictor's than that of Trajectron++.

Remarkably, the handcrafted RH predictor with a very simple rule hierarchy (Fig. 2(c)) has a performance comparable to Trajectron++; in fact, MPF prefers RH predictor over Trajectron++, as indicated by the average belief for the RH predictor being greater than 0.5 across all datasets in Table 4 provided in Appendix B.4. RH predictor has the best mean ADE and mean FDE among all predictors, but has a higher mean minADE and mean minFDE. The violin plots in Fig. 4 plot the spread of ADE, FDE, minADE, and minFDE across all scenes in each dataset. In Fig 4 we observe that Trajectron++ yields metrics which follow a unimodal distribution smoothly tapering to the tail. In contrast, the RH predictor performs well on a majority of scenes (a large mode near lower metric values), but also has a distinct mode with poor performance. This behavior can be attributed the RH predictor's limited ability to reason about multimodality in the agent's goal. As an example, in Fig. 3, the RH planner assumes the agent aims to follow the lane corresponding to the black dashed curve which turns right. However, in this case there are other plausible options as well, including the ground truth in this scenario where the agent turns to the left, shown in the pink curve. By switching between the Trajectron++ and RH predictors, we are able to take advantage of the RH predictor's superior performance when it predicts the correct lane line to follow and Trajectron++'s better performance when the RH predictor gets the desired lane line wrong. The improvement by MPF can be visually observed in Fig. 4 where the distribution for MPF mitigates the second mode of poor per-

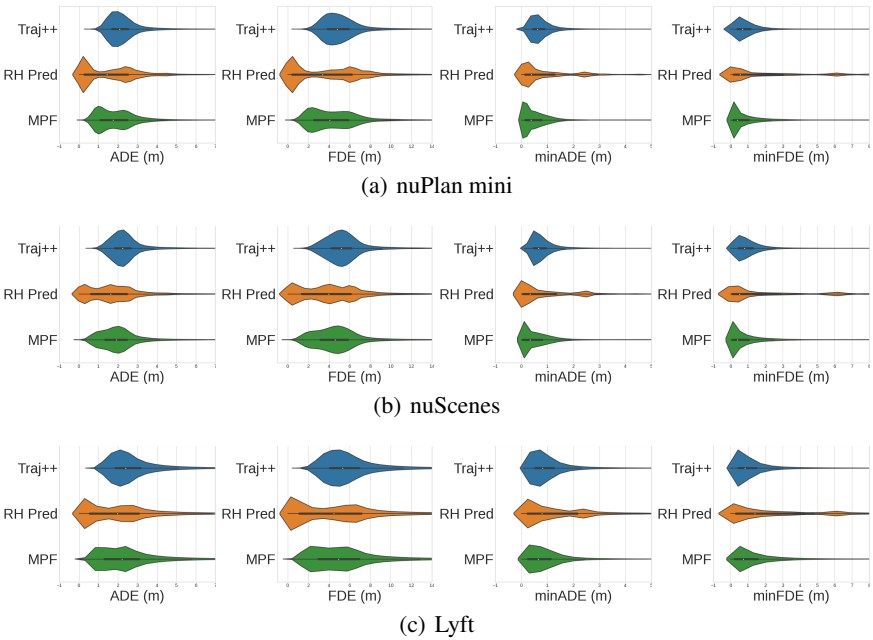

Figure 4: Violin plots for visualizing the spread of the metrics on the datasets for all three predictors.

formance by the RH predictor and shifts the entire distribution towards lower metric values relative to Trajectron++. Furthermore, we observe that our $\eta$-generalized Bayes filter approach performs better than the thresholded-likelihood approach in [23]; more details are provided in Appendix C.

**Runtime Considerations.** The average time for MPF to predict for a single agent at a given time-step is $\approx 0.08665$ seconds. Of this, the bulk of the time is spent in querying the standalone predictors and a very minuscule amount of time is spent on the fuser: on average, $0.06994$ seconds are spent on querying trajectories from the rule hierarchy predictor, $0.01671$ seconds from Trajectron++, and $0.013$ milliseconds on the belief update and re-sampling from the fuser. The speed of MPF can be attributed to the high degree of parallelization on the GPU. There is scope for further computational speed improvement by parallelizing the querying of the standalone predictors.

## 5    Conclusion, Limitations, and Future Work

**Conclusion.** In this work, we demonstrated that a cost function constructed by encoding traffic code in the form of a rule-hierarchy can be leveraged to produce competitive future predictions for vehicles in traffic. Furthermore, we proposed MPF, a method to fuse the predictions from the rule-hierarchy planner with predictions from a learned trajectory forecaster to yield a prediction framework which plays to the complementary strengths of each model. We evaluated the fusion approach on real-world trajectory prediction datasets and demonstrated the yields improved predictions relative to a learned baseline across a suite of metrics.

**Limitations.** Our approach has two main limitations. The first limitation is the additional computational expense of querying an ensemble of predictors as compared to running a single predictor. This challenge can be mitigated by parallelizing the predictor querying; however, it still involves some parallelization overhead and greater memory requirement on the GPU. The second limitation of MPF is the lack of multimodal behavior exhibited by the RH predictor, as discussed in Fig. 3. Finally, we remark that MPF shares the limitation of all agent-centric predictors: the need to parallelize predictions queries across all agents to ensure feasible computational run times.

**Future Work.** This work provides various exciting opportunities for future work. On the practical front, we will explore methods, such as [21], for introducing better multimodal behaviors for the planner in the rule hierarchy predictor, perform a closed-loop evaluation of MPF, parallelize predictions for all agents in a scene, and develop a more detailed rule hierarchy. On the theoretical front, we will explore the use of online learning techniques that provide convergence guarantees for the belief distribution.

**Acknowledgments**

We thank Boris Ivanovic for helpful discussions on trajectory prediction and Yuxiao Chen for his help with spline trajectory tree generation and lane planning.

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

## A  Algorithm for MPF

The algorithm for MPF is provided in Alg. 1. We use the agent-centric dataloader generated by `trajdata` [19] to access various prediction datasets ([32, 18, 17]) in a unified representation. In lines 15-16, trajectories from the standalone predictors are sampled, and in lines 8-13, the belief update discussed in Section 3.3.2 is performed.

---

**Algorithm 1** Multi-Predictor Fusion (MPF)

---

1: **Input:** `learn_predictor`, `rule_predictor`, `dataloader` (agent-centric)
2: **Output:** `pred_MPF` ← [] (list of MPF predictions)
3: **Hyperparameters:** number of trajectory samples per scene $N$, prediction horizon $T$, Bayes learning rate $\eta$, convex combination factor with prior $\gamma$ in (7), belief prior $b_0$.
4: **for** episode in dataloader **do**
5:   $t \leftarrow 0, b \leftarrow b_0$
6:   `learn_predictor_history`← $\emptyset$, `rule_predictor_history`← $\emptyset$,
7:   **for** scene in episode **do**
8:    **if** $t > 0$ **then**
9:     $x_t \leftarrow$ `observe_current_agent_state()`
10:     `likelihood_learn` ← `get_likelihood(learn_predictor_history`, $x_t$)
11:     `likelihood_rule` ← `get_likelihood(rule_predictor_history`, $x_t$)
12:     $\alpha \leftarrow$ (`likelihood_learn`$*b^l$)/(`likelihood_rule`$*b^r$)
13:     `new_belief` $\leftarrow \begin{bmatrix} \frac{\alpha^\eta}{1+\alpha^\eta} \\ \frac{1}{1+\alpha^\eta} \end{bmatrix}$
14:     $b \leftarrow (1-\gamma)$`new_belief` $+ \gamma b_0$
15:    **end if**
16:    $\{x^l_{t+1:t+T,i}\}^N_{i=1} \leftarrow$ `learn_predictor.sample(scene`, $N,T$)
17:    $\{x^r_{t+1:t+T,i}\}^N_{i=1} \leftarrow$ `rule_predictor.sample(scene`, $N,T$)
18:    `learn_predictor_history` $\leftarrow \{x^l_{t+1,i}\}^N_{i=1}$
19:    `rule_predictor_history` $\leftarrow \{x^r_{t+1,i}\}^N_{i=1}$
20:    `pred_MPF.append(belief_sampler(`$\{x^l_{t+1:t+T,i}\}^N_{i=1}, \{x^r_{t+1:t+T,i}\}^N_{i=1}, b$)
21:    $t+ = 1$
22:   **end for**
23: **end for**
24: **return** `pred_MPF`

---

## B  Further Details on Experimental Evaluation

### B.1  Hyperparameters

In our experimental evaluation we set the hyperparameters in line 3 of Alg. 1 as $N = 20$, $T = 8$, $\eta = 0.1$, $\gamma = 0.1$, and an uninformed prior $b_0 = [0.5, 0.5]$. The time $dt$ between two timesteps is 0.5 seconds.

### B.2  Dataset Details

The number of scenes and episodes per dataset are provided in Table 2 below:

| Dataset | Scenes | Episodes |
|---|---|---|
| nuPlan-mini val | 118030 | 6814 |
| nuScenes val | 66620 | 4140 |
| Lyft val 10% | 744448 | 49196 |

Table 2: Dataset Details

### B.3  $\eta$ and $\gamma$ study

We first set $\eta = 0.1$ and vary $\gamma$ in (7). The mean difference from best (MDB) for MPF, compared against Trajectron++ and RH Predictor, for different choices of $\gamma$ are reported on the left-hand side of Table 3. As can be noted from the last row, the performance of MPF is not very sensitive to the choice of $\gamma$. Then we choose the smallest best performing $\gamma$ (i.e., 0.1) and sweep across four values

of $\eta$. The results for that are reported on the right-hand side of Table 3. With $\eta$ we see a clear trend that smaller $\eta$ perform better than larger $\eta$ when MDB is averaged across all datasets. A large $\eta$ updates the belief more aggressively based on the past evidence, potentially making more mistakes compared to a smaller $\eta$ that results in more cautious belief updates.

| Dataset | MDB (%) @$\eta = 0.1$ | | | | MDB (%) @$\gamma = 0.1$ | | | |
|---|---|---|---|---|---|---|---|---|
| | @$\gamma = 0.05$ | @$\gamma = 0.1$ | @$\gamma = 0.15$ | @$\gamma = 0.20$ | @$\eta = 0.1$ | @$\eta = 0.4$ | @$\eta = 0.7$ | @$\eta = 1.0$ |
| NuPlan-mini | 4.31 | 4.36 | 4.41 | 4.46 | 4.36 | 4.29 | 4.18 | 3.72 |
| NuScenes | 5.19 | 5.17 | 5.16 | 5.16 | 5.17 | 5.23 | 5.42 | 6.45 |
| Lyft | 4.32 | 4.28 | 4.24 | 4.21 | 4.28 | 4.53 | 5.18 | 8.68 |
| Avg. MDB on Datasets | 4.61 | **4.60** | **4.60** | 4.61 | **4.60** | 4.68 | 4.93 | 6.28 |

Table 3: MDB for different $\eta$ and $\gamma$ in Bayes update.

## B.4 Belief statistics

For the MPF results presented in Table 1, we additionally provide statistics on the belief distribution between the RH predictor and Trajectron++ in Table 4. We observe that the mean belief for RH predictor is greater than $0.5$, indicating that on-average, MPF relies more on RH predictor than Trajectron++ across all datasets. We note that the standard deviation for the belief of both predictors is always the same because they are related by $b^l = 1 - b^r$, where $b^l$ and $b^r$ are the belief for Trajectron++ and RH predictor, respectively.

| Dataset | Predictor | Mean | Std | Min | Max |
|---|---|---|---|---|---|
| NuPlan-mini | RH Pred | 0.53 | 0.06 | 0.26 | 0.95 |
| | Traj++ | 0.47 | 0.06 | 0.05 | 0.74 |
| NuScenes | RH Pred | 0.54 | 0.03 | 0.07 | 0.92 |
| | Traj++ | 0.46 | 0.03 | 0.08 | 0.93 |
| Lyft | RH Pred | 0.54 | 0.05 | 0.05 | 0.94 |
| | Traj++ | 0.46 | 0.05 | 0.06 | 0.95 |

Table 4: Belief statistics for MPF results in Table 1.

## C  Comparison with [23]

In this section we compare the performance of our $\eta$-generalized Bayes filter used in MPF with the thresholded-likelihood approach in [23, Section II-F]. We compute the likelihood of the observation given Trajectron++, i.e., $\mathbb{P}(x_{t+1} \mid y_{t-H:t}, m, l)$ and switch to the RH predictor when it dips below a threshold $\nu$ as follows:

$$\begin{cases} \text{RH Predictor,} & \mathbb{P}(x_{t+1} \mid y_{t-H:t}, m, l) < \nu, \\ \text{Trajectron++,} & \text{otherwise} \end{cases} \qquad (8)$$

Since it is not clear what the best threshold should be, we sweep $\nu$ across 0.1, 0.2, 0.3, 0.4, and 0.5 and report all of them in Table 5. To further mimic the approach in [23], we use the same historical buffer window (i.e., time delay in switching) of 3-seconds; [23] uses 30-timesteps which corresponds to a 3-second window according to their dataset.

In Table 5, we notice that [23] serves as a strong baseline which results in a mixed model that does better than Trajectron++ and RH predictor on many metrics (compare with Table 1). However, overall MPF outperforms the switching approach of [23], as summarized by the lowest MDB[3] for all three datasets in Table 5. The reason behind this is MPF's ability to mix trajectories at every time-step which allows it to rapidly respond to performance degradation by a predictor. MPF is able to achieve this because it maintains a belief distribution over the standalone predictors, unlike [23].

---

[3]In Table 5, MDB is computed across six predictors: five predictors based on [23] with varying $\nu$ and MPF.

**NuPlan mini**

| Predictor | Mean (m) | | | | CVaR$_{0.1}$ (m) | | | | Mean Diff From Best (MDB) (%) |
|---|---|---|---|---|---|---|---|---|---|
| | ADE | FDE | minADE | minFDE | ADE | FDE | minADE | minFDE | |
| $\nu = 0.1$ | $2.28 \pm 0.01$ | $5.18 \pm 0.01$ | $0.78 \pm 0.00$ | $0.97 \pm 0.01$ | $4.51 \pm 0.04$ | $9.88 \pm 0.06$ | $2.23 \pm 0.04$ | $3.44 \pm 0.06$ | 16.42 |
| $\nu = 0.2$ | $2.27 \pm 0.01$ | $5.16 \pm 0.01$ | $0.77 \pm 0.00$ | $0.97 \pm 0.01$ | $4.42 \pm 0.04$ | $9.77 \pm 0.06$ | $2.17 \pm 0.04$ | $3.42 \pm 0.06$ | 15.26 |
| $\nu = 0.3$ | $2.26 \pm 0.01$ | $5.16 \pm 0.01$ | $0.77 \pm 0.00$ | $0.97 \pm 0.01$ | $4.38 \pm 0.04$ | $\mathbf{9.74 \pm 0.06}$ | $2.14 \pm 0.03$ | $\mathbf{3.41 \pm 0.06}$ | 14.68 |
| $\nu = 0.4$ | $2.26 \pm 0.01$ | $5.16 \pm 0.01$ | $0.77 \pm 0.00$ | $0.97 \pm 0.01$ | $4.36 \pm 0.04$ | $\mathbf{9.74 \pm 0.06}$ | $2.14 \pm 0.04$ | $3.46 \pm 0.06$ | 14.80 |
| $\nu = 0.5$ | $2.26 \pm 0.01$ | $5.16 \pm 0.01$ | $0.77 \pm 0.00$ | $0.98 \pm 0.01$ | $4.37 \pm 0.04$ | $9.78 \pm 0.07$ | $2.15 \pm 0.04$ | $3.55 \pm 0.06$ | 15.53 |
| MPF (ours) | $\mathbf{1.95 \pm 0.01}$ | $\mathbf{4.53 \pm 0.02}$ | $\mathbf{0.54 \pm 0.00}$ | $\mathbf{0.80 \pm 0.01}$ | $\mathbf{4.34 \pm 0.02}$ | $10.04 \pm 0.06$ | $\mathbf{1.74 \pm 0.01}$ | $3.41 \pm 0.03$ | $\mathbf{0.40}$ |

**NuScenes**

| Predictor | Mean (m) | | | | CVaR$_{0.1}$ (m) | | | | Mean Diff From Best (MDB) (%) |
|---|---|---|---|---|---|---|---|---|---|
| | ADE | FDE | minADE | minFDE | ADE | FDE | minADE | minFDE | |
| $\nu = 0.1$ | $2.40 \pm 0.01$ | $5.48 \pm 0.02$ | $0.81 \pm 0.00$ | $1.06 \pm 0.01$ | $\mathbf{4.56 \pm 0.04}$ | $\mathbf{10.07 \pm 0.07}$ | $2.16 \pm 0.03$ | $\mathbf{3.49 \pm 0.05}$ | 14.07 |
| $\nu = 0.2$ | $2.40 \pm 0.01$ | $5.48 \pm 0.02$ | $0.81 \pm 0.00$ | $1.06 \pm 0.01$ | $\mathbf{4.56 \pm 0.04}$ | $10.08 \pm 0.07$ | $2.17 \pm 0.03$ | $3.50 \pm 0.05$ | 14.17 |
| $\nu = 0.3$ | $2.40 \pm 0.01$ | $5.48 \pm 0.02$ | $0.82 \pm 0.00$ | $1.07 \pm 0.01$ | $4.57 \pm 0.04$ | $10.09 \pm 0.07$ | $2.18 \pm 0.03$ | $3.53 \pm 0.06$ | 14.42 |
| $\nu = 0.4$ | $2.40 \pm 0.01$ | $5.48 \pm 0.02$ | $0.82 \pm 0.00$ | $1.07 \pm 0.01$ | $4.57 \pm 0.04$ | $10.11 \pm 0.07$ | $2.20 \pm 0.03$ | $3.60 \pm 0.06$ | 15.03 |
| $\nu = 0.5$ | $2.41 \pm 0.01$ | $5.50 \pm 0.02$ | $0.83 \pm 0.01$ | $1.12 \pm 0.01$ | $4.64 \pm 0.04$ | $10.26 \pm 0.08$ | $2.31 \pm 0.03$ | $3.94 \pm 0.07$ | 18.22 |
| MPF (ours) | $\mathbf{2.09 \pm 0.01}$ | $\mathbf{4.85 \pm 0.02}$ | $\mathbf{0.54 \pm 0.01}$ | $\mathbf{0.83 \pm 0.01}$ | $4.70 \pm 0.04$ | $10.34 \pm 0.08$ | $\mathbf{2.04 \pm 0.03}$ | $3.73 \pm 0.05$ | $\mathbf{1.58}$ |

**Lyft**

| Predictor | Mean (m) | | | | CVaR$_{0.1}$ (m) | | | | Mean Diff From Best (MDB) (%) |
|---|---|---|---|---|---|---|---|---|---|
| | ADE | FDE | minADE | minFDE | ADE | FDE | minADE | minFDE | |
| $\nu = 0.1$ | $2.78 \pm 0.00$ | $6.03 \pm 0.01$ | $1.05 \pm 0.00$ | $1.30 \pm 0.00$ | $\mathbf{6.23 \pm 0.01}$ | $\mathbf{13.19 \pm 0.03}$ | $3.12 \pm 0.01$ | $\mathbf{5.07 \pm 0.02}$ | 6.46 |
| $\nu = 0.2$ | $2.78 \pm 0.00$ | $6.03 \pm 0.01$ | $1.05 \pm 0.00$ | $1.31 \pm 0.00$ | $\mathbf{6.23 \pm 0.01}$ | $13.21 \pm 0.03$ | $3.14 \pm 0.01$ | $5.13 \pm 0.02$ | 6.75 |
| $\nu = 0.3$ | $2.78 \pm 0.00$ | $6.04 \pm 0.01$ | $1.06 \pm 0.00$ | $1.33 \pm 0.00$ | $\mathbf{6.23 \pm 0.01}$ | $13.28 \pm 0.03$ | $3.19 \pm 0.01$ | $5.31 \pm 0.03$ | 7.78 |
| $\nu = 0.4$ | $2.78 \pm 0.00$ | $6.06 \pm 0.01$ | $1.07 \pm 0.00$ | $1.40 \pm 0.01$ | $6.33 \pm 0.02$ | $13.57 \pm 0.03$ | $3.38 \pm 0.01$ | $5.88 \pm 0.03$ | 11.47 |
| $\nu = 0.5$ | $2.80 \pm 0.00$ | $6.14 \pm 0.01$ | $1.12 \pm 0.00$ | $1.57 \pm 0.01$ | $6.59 \pm 0.02$ | $14.31 \pm 0.04$ | $3.83 \pm 0.02$ | $7.22 \pm 0.05$ | 20.66 |
| MPF (ours) | $\mathbf{2.55 \pm 0.00}$ | $\mathbf{5.69 \pm 0.01}$ | $\mathbf{0.85 \pm 0.00}$ | $\mathbf{1.25 \pm 0.00}$ | $6.63 \pm 0.02$ | $14.74 \pm 0.03$ | $\mathbf{2.87 \pm 0.01}$ | $5.18 \pm 0.02$ | $\mathbf{2.53}$ |

Table 5: Results comparison between MPF and the Bayesian failure monitoring approach in [23, Section II-F] for 4-second prediction horizon with $N = 20$ rounded to the nearest second decimal.

