# OpenReview forum: "Multi-Predictor Fusion: Combining Learning-based and Rule-based Trajectory Predictors"
_robot-learning.org/CoRL/2023/Conference — CoRL 2023 Poster_

### Official Review · Reviewer_VeXK · 2023-07-19

**Confidence:** 3
**Originality:** Very Good
**Technical Quality:** Very Good
**Clarity Of Presentation:** Excellent
**Impact:** 3

**Recommendation:**

Weak Accept: I recommend accepting the paper, but will not argue for my recommendation if the majority of other reviewers have a different opinion.

**Review:**

I found that the paper is well developed, with clarity and sufficient contents, to describe the concept and to illustrate the performance of the proposed algorithm The topic is of great interest in the community, and the contributions of the paper are significant to the field.

**Quality Of The Limitations Section:**

Limitations are addressed clearly

**Questions For Rebuttal:**

None

**Robotics Focus:**

Highly relevant to robotics but no hardware experiments

**Summary Of Paper:**

An algorithm called multi-predictor fusion (MPF) that augments the performance of learning-based predictors is presented. MPF probabilistically combines learning- and rule-based predictors, by essentially mixing trajectories from both standalone predictors in accordance with a belief distribution that reflects the online performance of each predictor. It is shown in the paper that MPF outperforms the two standalone predictors on various metrics and delivers the most consistent performance.

**Summary Of Recommendation:**

I have some background in the field and feel that the paper is of significant contribution to the filed and community.

---

### Official Review · Reviewer_4qup · 2023-07-19

**Confidence:** 5
**Originality:** Good
**Technical Quality:** Good
**Clarity Of Presentation:** Good
**Impact:** 4

**Recommendation:**

Weak Accept: I recommend accepting the paper, but will not argue for my recommendation if the majority of other reviewers have a different opinion.

**Review:**

Strengths:

* The paper is well-written and easy to follow.
* The proposed multi-predictor fusion module can be used with any off-the-shelf predictor, making it easily applicable to a variety of scenarios.
* The idea of combining learning-based and rule-based motion predictors is beneficial to real-world robotics applications.

Weaknesses:

* Computation expense may be a key problem to solve for real-world robotics applications, as the proposed method may require significant computational resources.
* The RH predictor does not exhibit multimodal behavior, which may limit its effectiveness in scenarios where multiple outcomes are possible.
* The paper would be better to address how the rule-based motion predictor would perform in scenarios where an HD map is not available, such as in a large intersection. This limitation should be discussed in the paper.


**Quality Of The Limitations Section:**

Limitations are addressed clearly

**Questions For Rebuttal:**

* The literature review could benefit from more discussions on fusing learned and rule-based trajectory prediction to provide more context for the proposed method.
* The paper should include more quantitative results on different datasets to demonstrate the effectiveness and generalization of the proposed method.
* The computation expense of the proposed method may be a severe problem when applied to multi-agent motion prediction problems, and this limitation should be discussed in the paper.

**Robotics Focus:**

Highly relevant to robotics but no hardware experiments

**Summary Of Paper:**

 The paper presents an algorithm called multi-predictor fusion (MPF) that enhances the performance of learning-based trajectory predictors for the safe and efficient planning of autonomous vehicles in highly interactive traffic scenarios. MPF combines learning- and rule-based predictors by mixing trajectories from both predictors in accordance with a belief distribution that reflects the online performance of each predictor. The proposed algorithm is shown to outperform the two standalone predictors on various metrics and delivers the most consistent performance in the results. The authors demonstrate that MPF augments the performance of learning-based predictors by imbuing them with motion planners that are tasked with satisfying logic-based rules.

**Summary Of Recommendation:**

Based on the strengths highlighted in the review, it appears that the paper proposes an interesting idea that may have an impact on real-world robotics applications. Therefore, I recommend accepting the paper for publication in CoRL. However, the weaknesses and concerns raised in the review should be addressed in the final version of the paper.

---

### Official Review · Reviewer_oqEg · 2023-07-20

**Confidence:** 4
**Originality:** Good
**Technical Quality:** Good
**Clarity Of Presentation:** Very Good
**Impact:** 3

**Recommendation:**

Weak Accept: I recommend accepting the paper, but will not argue for my recommendation if the majority of other reviewers have a different opinion.

**Review:**

**Strength**: The proposed rule-based planner based on rule hierarchy is interesting. It is quite impressive that it achieved performance comparable to Trajectron++.

**Weakness**:  A very relevant and crucially important prior work [1] was missing in the related work. In [1], various methods to fuse learning- and model-based prediction models were proposed and evaluated, including a recursive Bayesian inference scheme that is very similar to the one proposed in this work. Thus, I would not consider the idea of predictor fusion and the proposed fusion scheme, which are claimed as the main novelty of this work, as novel and significant. The authors should compare their work with [1] and justify the novelty of their work during rebuttal.

[1] Sun, L., X. Jia, and A. Dragan. "On Complementing End-To-End Human Behavior Predictors with Planning." Robotics science and systems (RSS). 2021.

**Quality Of The Limitations Section:**

Additional details required

**Questions For Rebuttal:**

1. The proposed recursive Bayesian inference scheme depends on both $\gamma$ and $\eta$. While the authors reported an ablation study on different values of $\eta$ in Appendix B.3, there should be an ablation study on the effect of different $\gamma$ values on the fusion performance. Besides, the authors claimed, "The performance of MPF is not very sensitive to the choice of $\eta$". It is the case in NuPlan-mini and NuScenes, but the performance in Lyft has relatively large variation given different $eta$. Could the authors give some insights or hypotheses on it?

2. In the rule-based predictor, it is assumed that "All vehicles, other than the one for which we are predicting, are assumed to move with a constant velocity over the prediction horizon for computing the reward". It is certainly a very big assumption. The authors should provide justification for this assumption. Also, does this assumption affect the prediction performance of RH prediction? Can any of those cases where RH has large prediction errors be explained by the violation of this assumption?

**Robotics Focus:**

Highly relevant to robotics but no hardware experiments

**Summary Of Paper:**

This work proposed a rule-based trajectory predictor using rule hierarchies that achieved comparable performance to Trajectron++. Then the authors proposed the multi-predictor fusion (MPF) algorithm, which fuses the learning- and rule-based predictors using recursive Bayesian multi-modal filtering. It was shown that MPF achieved more consistent performance across different prediction metrics than the learning- and rule-based predictors.

**Summary Of Recommendation:**

The most critical concern I have regarding this paper is the novelty and significance of the predictor fusion scheme. Thus, I lean towards rejecting this paper, given its current form. I am happy to raise my score if the authors can properly address my concerns during rebuttal.

---

### Author Response · Authors · 2023-08-10
**Summary of Changes to the Paper**

We appreciate the time the reviewers spent on the evaluation of our work and thank them for their valuable feedback. We have responded to the comments provided by the reviewers and incorporated their feedback in an updated draft of the paper (including the appendix) which is shared in the response to each reviewer. All changes in the paper are indicated in blue color. The main changes are:

* We have included an experimental comparison with the Bayesian approach in [1] in the appendix of the paper. The comparisons show that [1] serves as a strong baseline; however, the method proposed in our paper outperforms the method in [1].
* We have included an additional ablation study with $\gamma$ and also statistics on the belief distribution in the appendix of our paper.
* We have updated the literature review, limitations, and future work in accordance with the reviewers’ comments.

**References**

[1] Sun, L., X. Jia, and A. Dragan. “On Complementing End-To-End Human Behavior Predictors with Planning.” Robotics science and systems (RSS). 2021.

---

### Decision · Program_Chairs · 2023-08-30

**Decision:**

Accept (Poster)

**Comment:**

The paper presents a very interesting idea for a tile-based method for trajectory prediction that is comparable to trajectron++. All reviewers agreed on the strengths of the paper on the multi-predictor fusion module that can be used with any off-the-shelf predictor, making it a flexible solution with a possible impact on real-world robotics applications. The updated version of the paper has also integrated well most of the reviewers feedback. I would suggest that the authors rearrange some experiments in their camera ready version to better highlight their contributions and findings.